Proceedings of the 6th Symposium on Advances in Approximate Bayesian Inference, 2024 1–19

# PAC-Bayesian Soft Actor-Critic Learning

**Bahareh Tasdighi**                    Tasdighi@imada.sdu.dk

**Abdullah Akgül**                    akgul@imada.sdu.dk

**Manuel Haussmann**                    haussmann@imada.sdu.dk

**Kenny Kazimirzak Brink**                    kebri18@student.sdu.dk

**Melih Kandemir**                    kandemir@imada.sdu.dk

*University of Southern Denmark*

## Abstract

Actor-critic algorithms address the dual goals of reinforcement learning (RL), policy evaluation and improvement via two separate function approximators. The practicality of this approach comes at the expense of training instability, caused mainly by the destructive effect of the approximation errors of the critic on the actor. We tackle this bottleneck by employing an existing Probably Approximately Correct (PAC) Bayesian bound for the first time as the critic training objective of the Soft Actor-Critic (SAC) algorithm. We further demonstrate that online learning performance improves significantly when a stochastic actor explores multiple futures by critic-guided random search. We observe our resulting algorithm to compare favorably against the state-of-the-art SAC implementation on multiple classical control and locomotion tasks in terms of both sample efficiency and regret.

## 1. Introduction

The process of searching for an optimal policy to govern a Markov Decision Process (MDP) involves solving two sub-problems: (i) policy evaluation and (ii) policy improvement (Bertsekas and Tsitsiklis, 1996). The policy evaluation step identifies a function that maps a state to its value, i.e., the total expected reward the agent will collect by following a predetermined policy. The policy improvement step updates the policy parameters such that the states with larger values are visited more frequently. Modern actor-critic methods (Peters et al., 2010; Schulman et al., 2015; Lillicrap et al., 2016; Schulman et al., 2017; Haarnoja et al., 2018) address the two-step nature of policy search by allocating a separate neural network for each step, a *critic* network for policy evaluation and an *actor* network for policy improvement. It is often straightforward to achieve policy improvement by taking gradient-ascent steps on actor parameters to maximize the critic's output. However, as the training objective of the actor-network, the accuracy of the critic sets a severe bottleneck on the success of the eventual policy search algorithm at the target task. The state-of-the-art attempts to overcome this bottleneck using copies of critic networks as Bellman target estimators whose parameters are updated with a time lag (Lillicrap et al., 2016) and using the minimum of two critics for policy evaluation to tackle overestimation due to the Jensen gap (Fujimoto et al., 2018; Haarnoja et al., 2018).

Probably Approximately Correct (PAC) Bayesian theory (Shawe-Taylor and Williamson, 1997; McAllester, 1999) develops analytical statements about the worst-case generalization performances of Gibbs predictors that hold with high probability (Alquier, 2021). The theory assumes the predictors to follow a posterior measure tunable to observations

while maintaining similarity to a desired prior measure. Although this prior and posterior nomenclature resembles that of classical Bayesian theory, as, e.g., the two distributions are not constraint with respect to each other by a given likelihood. Rather it forms part of a growing literature known as generalized Bayesian inference (Guedj, 2019; Matsubara et al., 2022). In addition to being in widespread use for deriving analytical guarantees for model performance, PAC Bayesian bounds are useful for developing training objectives with learning-theoretic justifications in supervised learning (Dziugaite and Roy, 2017) and dynamical systems modeling (Haußmann et al., 2021). Although PAC Bayesian bounds offer useful tools for deriving conservative losses, which have many applications in reinforcement learning, their potential as training objectives has remained relatively unexplored to date.

In this paper, we demonstrate how PAC Bayesian bounds can improve the performance of modern actor-critic algorithms when used for robust policy evaluation. We adapt an existing PAC Bayesian bound (Fard et al., 2012) developed earlier for transfer learning for the first time to train the critic network of a SAC algorithm (Haarnoja et al., 2018). We discover the following outcomes:

- A PAC Bayesian bound can predict the worst-case critic performance with high probability. When used as a training objective, it overcomes the value overestimation problem of approximate value iteration (Van Hasselt et al., 2016) even with a single critic and brings about an improved regret profile in the online learning setting.

- The randomized critic expedites policy improvement when used as a guide for optimal action search via multiple shooting. It does so by sampling multiple one-step-ahead imaginary futures from the randomized critic and actor and chooses the action that gives the highest sampled state-action value.

Based on these two outcomes, we propose a novel algorithm called *PAC Bayes for Soft Actor-Critic (PAC4SAC)*, which delivers consistent performance gains over not only the vanilla SAC algorithm but also alternative approaches to robust online reinforcement learning. We observe our PAC4SAC to solve four continuous control tasks with varying levels of difficulty in fewer environment interactions and smaller cumulative regret than its counterparts. Figure 1 illustrates the main idea of the proposed algorithm.

## 2. Background

**Maximum entropy reinforcement learning.** We model a learning agent and its environment as a Markov Decision Process (MDP), given by a tuple $(\mathcal{S}, \mathcal{A}, p, p_0, r, \gamma)$ comprising of a $d_s$-dimensional continuous state space $\mathcal{S} := \{s \in \mathbb{R}^{d_s}\}$, a $d_a$-dimensional continuous action space $\mathcal{A} := \{a \in \mathbb{R}^{d_a}\}$, a reward function with bounded range $r : \mathcal{S} \times \mathcal{A} \to [0, R_{\max}]$, a reward discount factor $\gamma \in [0, 1)$, a state transition distribution $s' \sim P(\cdot|s, a)$, and an initial state distribution $p_0(s_0)$. We denote the visitation density function for state $s'$ recursively as $p_\pi(s') = \mathbb{E}_{s \sim p_\pi, \pi}[P(s'|s, a)]$, where $\mathbb{E}_{x \sim \mu}[f(x)]$ is the expectation of a function $f$ measurable by $\mu$. The corresponding variance is $\text{var}_{x \sim \mu}[f(x)]$. The terminal condition of the recursion is defined as $p_\pi(s_0) := p_0(s_0)$. The goal of maximum entropy online reinforcement learning is to find a policy distribution $\pi(a|s)$ such that both its entropy and the expected

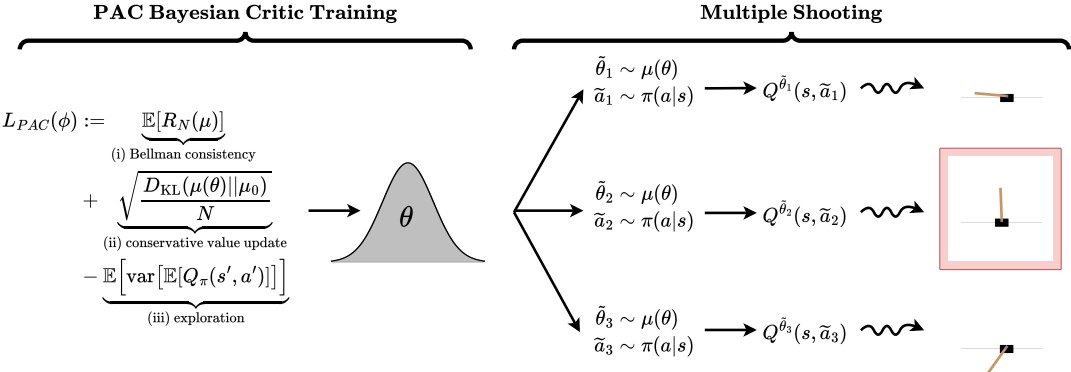

Figure 1: Our novel *Probably Approximately Correct Bayes for Soft Actor Critic (PAC4SAC)* algorithm trains a critic with random parameters $\theta$ for the first time using a PAC Bayesian bound as its training objective. The random critic enables effective random optimal action search when used as a guide for a stochastic policy. The resulting algorithm solves online reinforcement learning tasks with fewer environment interactions and smaller cumulative regret than its counterparts.

cumulative reward are maximized after a minimum number of environment interactions,

$$\pi_* = \arg\max_\pi \sum_{i=0}^\infty \mathbb{E}_{s\sim p_\pi, a\sim\pi} \left[ \gamma^i r(s,a) - \alpha \log \pi(a|s) \right], \tag{1}$$

where the coefficient $\alpha > 0$ is a parameter that regularizes the importance of the policy entropy. We study the model-free case where the true state transition distribution $P$ and the true reward function $r(s,a)$ are unknown to the agent and are integrated out during training from the observed state, action, reward, and next state tuples $(s, a, r, s')$.

**Actor-critic learning.** We define the true value of a state-action pair under policy $\pi$, referred to as the *actor*, at time step $t$ of an interaction round with an environment as

$$Q_\pi(s,a) := r(s,a) + \sum_{t=0}^\infty \mathbb{E}_{s'\sim p_\pi, a'\sim\pi} \left[ \gamma^t r - \alpha \log \pi(a'|s') \right],$$

using $r$ for the case when $r(s,a)$ is observed. As the true value function is unknown to us, we approximate it by a trainable function $Q$, e.g., a neural network, referred to as the *critic*. One incurs a Bellman error if there exists a pair $(s,a) \in \mathcal{S} \times \mathcal{A}$ such that $Q_\pi(s,a) \neq Q(s,a)$, which can be quantified as the squared error

$$L_N(\theta) = \sum_{(s,a,r,s')\in D_N} \left( r + \gamma Q(s',a') - Q(s,a) \right)^2,$$

for a sample set $D_N$ containing $N$ tuples $(s, a, r, s')$ known as a *replay buffer*. The *Soft Actor-Critic (SAC)* algorithm (Haarnoja et al., 2018) is trained by alternating between policy

evaluation $\arg\min_\theta L_N(\theta)$ which minimizes the Bellman error, and policy improvement

$$\arg\max_\pi \frac{1}{N} \sum_{(s,a)\in D_N} \mathbb{E}_{a_i\sim\pi}\left[Q(s,a) - \alpha\log\pi(a_i|s_i)\right], \tag{2}$$

which is equivalent to minimizing the Kullback-Leibler (KL) divergence between the policy distribution and a Gibbs distribution derived from the value predictor

$$\arg\min_\pi \mathbb{E}_{s\in P}\left[D_{\mathrm{KL}}\left(\pi(\cdot|s)\left\|\frac{\exp(Q(\cdot|s)/\alpha)}{\int \exp(Q(\bar{a}|s)/\alpha)d\bar{a}}\right)\right],$$

where $\bar{a}$ relative to the probabilities of taking all other possible actions in the state $s$ and $D_{\mathrm{KL}}(\mu\|\mu') = \mathbb{E}_{x\sim\mu}\left[\log\mu(x) - \log\mu'(x)\right]$ for measures $\mu$ and $\mu'$.

**PAC Bayesian analysis.** Assume a prediction task from an input $x \in \mathcal{X}$ to output $y \in \mathcal{Y}$ with an unknown joint distribution $x, y \sim \mathcal{D}$, the performance of which is quantified by a bounded loss functional $L(h(x), y) : \mathcal{Y} \times \mathcal{Y} \to [0, L_{\max}]$, where $h$ is a prediction hypothesis. The main concern of statistical learning theory is to find the tightest possible bound for the risk functional $R(h) = \mathbb{E}_{(x,y)\sim\mathcal{D}}\left[L(h(x), y)\right]$ that holds with the highest possible probability based on its empirical estimate $R_N(h) = \frac{1}{N}\sum_{(x,y)\in D_N} L(h(x), y)$ for a data set $D_N$ of $N$ independent and identically distributed (i.i.d.) samples $(x, y)$ taken from a data distribution $\mathcal{D}$. Bounds that satisfy these desiderata, referred to as *Probably Approximately Correct (PAC)* (Kearns and Vazirani, 1994), follow the structure $\Pr[R(h) \le C(R_N(h), \delta)] \ge 1 - \delta$ where $C(\delta)$ is an analytical expression dependent on the empirical estimate $R_N$ and $\delta \in (0, 1)$ is a tolerance level. PAC Bayesian bounds (Shawe-Taylor and Williamson, 1997; McAllester, 1999) extend the PAC framework to the case where the hypothesis is assumed to follow a posterior distribution $h \sim \mu$. The measure $\mu_0$ denotes the prior distribution that represents domain knowledge or design choices to be imposed to the learning process. Differently from Bayesian inference, the relationship between prior and posterior distribution does not necessarily follow the Bayes theorem in the PAC Bayesian framework. It is sufficient that $\mu_0$ is chosen a priori, while $\mu$ may be fit to data. A PAC Bayesian bound has the structure

$$\Pr\left[\mathbb{E}_{h\sim\mu}\left[R(h)\right] \le C\big(\mathbb{E}_{h\sim\mu}\left[R_N(h)\right], \delta, D_{\mathrm{KL}}(\mu\|\mu_0)\big)\right] \ge 1 - \delta.$$

The key difference of this bound from a PAC bound is that it makes a statement about a distribution $q$ on the whole hypothesis space rather than a single hypothesis. Hence, it involves a complexity penalty at the scale of distributions $D_{\mathrm{KL}}(\mu\|\mu_0)$.

## 3. PAC Bayesian Soft Actor-Critic Learning

**Main problem.** The performance of online reinforcement learning algorithms is highly sensitive to the precision of the action-value function approximator, i.e., the critic. This sensitively causes poor performance and sample inefficiency in practice. The two key factors behind this are *(i) Bias in value estimation.* When the same function approximator is used for both prediction and target state estimation, the value of the target state is likely to be overestimated in Q-learning as a result of the approximation error. Even for additive

noise term $\epsilon$ with zero mean, it holds that $\mathbb{E}_\epsilon\left[\max_{a'}(Q(s',a')+\epsilon)\right] \geq \max_{a'} Q(s',a')$ (Thrun and Schwartz, 1993). The same bias has been shown to exist also in actor-critic algorithms (Fujimoto et al., 2018) and to accumulate throughout the training period, resulting in a risk of significant drop in online learning performance. Solutions such as using double critics results in an underestimation bias (Hasselt, 2010). *(ii) Catastrophic interference.* Updating $Q$ to reduce the Bellman error of a small group of states with poor value estimations affects also the other states, many of which may already have accurate value estimations (Pritzel et al., 2017). The common solution to mitigate these problems is Polyak averaging $Q \leftarrow (1-\tau)Q + \tau Q'$ for some $\tau \in (0,1)$.

> We claim that training a ***single*** randomized critic with a PAC-Bayesian generalization performance bound reduces the underestimation bias, while not suffering from catastrophic interference. We further claim that using all sources of randomization in the model for optimal action search brings additional performance boost.

### 3.1. Building a trainable PAC Bayes bound for SAC

We define our risk functional as

$$R(\mu) = \mathbb{E}_{Q\sim\mu}\left[\mathbb{E}_{s,s'\sim p_\pi, a\sim\pi}\left[(y-Q(s,a))^2\right]\right],$$

where $y = r + \gamma\mathbb{E}_{a'\sim\pi}\left[Q(s',a') - \log\pi(a'|s')\right]$ is the soft Bellman target.

$$R_N(\mu) = \frac{1}{N}\sum_{(s,a,r,s')\in D_N}\mathbb{E}_{Q\sim\mu}\left[(y-Q(s,a))^2\right],$$

forms the corresponding empirical estimate and is computed on a replay buffer of $N$ tuples of $(s,a,r,s')$ collected from the target environment. The only PAC Bayes bound available for this setting is by Fard et al. (2012). It was developed to evaluate a fixed policy executed only once on the target environment. Since this design choice brings about a strongly correlated random variable chain, the bound needs to account for this correlation. Define by $P_i(\cdot|s,a)$ the density function of the random variable $\Pr^i(s_{t+i} \in A|s_t, a_t), \forall A \in \mathcal{S}$ that quantifies the probability of an event $A$ taking place $i$ time steps after a reference time step $t$. The upper-triangular matrix $\Gamma_\pi^N = (\xi_{ij}) \in \mathbb{R}^{N\times N}$ with entries

$$\xi_{ij} = \max_{s,a,\tilde{s},\tilde{a}}\left|\left|\mathbb{E}_{a,\tilde{a}\sim\pi}\left[P^{j-i}(\cdot|s,a) - P^{j-i}(\cdot|\tilde{s},\tilde{a})\right]\right|\right|_1,$$

for $1 \leq i \leq j \leq N$, and zero otherwise, quantifies a measure of correlation of a random sequence $s_t, s_{t+1}, \ldots, s_{t+N}$ where $||\cdot||_1$ denotes the $L_1$ norm of the function inside the argument. Let us also denote $||f(x)||_p = \sqrt{\mathbb{E}_{x\sim p}\left[f(x)^2\right]}$ for some function $f$ with bounded range and a probability measure $p$. After being adapted to the maximum entropy setting and replacing all its constants by their actual values, Fard et al. (2012)'s bound becomes:

**Theorem 1** *For any posterior measure $\mu$ and any prior measure $\mu_0$ defined on the space of action-value functions $Q$ and any data set $N$ containing $(s,a,r,s')$ collected from a single*

execution of a fixed policy $\pi$, the following inequality holds with probability greater than $1-\delta$:

$$\mathbb{E}_{Q\sim\mu}\left[||Q_\pi - Q||^2_{p_\pi}\right] \leq \frac{1}{(1-\gamma)^2}\left(\mathbb{E}_{Q\sim\mu}\left[R_N(\mu)\right] + \sqrt{\frac{\log\left(\frac{N}{2R^2_{\max}||\Gamma^N_\pi||^2\delta}\right) + D_{KL}(\mu||\mu_0)}{\frac{N}{2R^2_{\max}||\Gamma^N_\pi||^2} - 1}}\right.$$

$$\left. - \mathbb{E}_{Q\sim\mu}\left[\text{var}_{s'\sim P}\left[\mathbb{E}_{a'\sim\pi}\left[\gamma Q(s',a') - \log\pi(a'|s')\right]\right]\right]\right).$$

where $||\Gamma^N_\pi||$ is the operator norm of the matrix in its argument.

> This bound faces several major shortcomings. (i) Its assumption on a single roll-out introduces a significant gap due to the $||\Gamma_N||^2$ factor on the denominator the square-root term. (ii) The value of this term is not known. (iii) For the bound to be valid, $N$, the length of a single episode, must be greater than $2R^2_{\max}||\Gamma_N||^2$, which is nearly impossible to satisfy in real-world applications.

We propose to construct an alternative bound that has similar qualitative properties of the bound above, but a significantly higher practical relevance. Our key insight is that the single-episode assumption is neither realistic nor necessary, as modern approaches to deep reinforcement learning maintain a large replay buffer that both increases $N$ dramatically and significantly decouples the correlation of the samples used in a single minibatch. We find it realistic enough to assume that the critic is trained with approximately i.i.d. $(s, a, r, s')$ tuples and build the PAC Bayes bound accordingly. Our starting point is the standard McAllester bound (McAllester, 1999)

$$\mathbb{E}_{Q\sim\mu}\left[R(\mu)\right] \leq \mathbb{E}_{Q\sim\mu}\left[R_N(\mu)\right] + \sqrt{\frac{\log(1/\delta) + D_{\text{KL}}(\mu||\mu_0)}{2N}}.$$

We use the following property suggested by Antos et al. (2008) and adopted by Fard et al. (2012) to relate the bound defined on the Bellman error to the value approximation error:

$$||Q_\pi - Q||^2_{p_\pi} \leq \frac{1}{(1-\gamma)^2}\left(\underbrace{||T_\pi Q - Q||^2_{p_\pi} + \mathbb{E}_{s\sim p_\pi}\left[\text{var}_{s'\sim P}\left[\mathbb{E}_{a'\sim\pi}\left[Q(s',a')\right]\right]\right]}_{R(\mu)}\right). \quad (3)$$

The final bound is then

$$\mathbb{E}_{Q\sim\mu}\left[R(\mu)\right] \leq \frac{1}{(1-\gamma)^2}\left(\mathbb{E}_{Q\sim\mu}\left[R_N(\mu)\right] + \sqrt{\frac{\log(1/\delta) + D_{\text{KL}}(\mu||\mu_0)}{2N}}\right.$$

$$\left. - \mathbb{E}_{Q\sim\mu}\left[\mathbb{E}_{s\sim p_\pi}\left[\text{var}_{s'\sim P}\left[\mathbb{E}_{a'\sim\pi}\left[Q(s',a')\right]\right]\right]\right]\right),$$

for any posterior measure $\mu$, prior measure $\mu_0$, and replay buffer $D_N$ with $N$ tuples of $(s, a, r, s')$ collected from arbitrarily many episodes. The right-hand side of this bound will get tighter as $\mu$ is fit to a given data sample $D_N$. Known as *PAC Bayesian Learning*, training machine learning models by minimizing PAC Bayesian bounds have shown remarkable

outcomes in deep learning (Dziugaite and Roy, 2017; Reeb et al., 2018) and attracted significant attention. We apply this idea for the first time to reinforcement learning and propose to train the critic network by solving the following optimization problem

$$\mu_* \leftarrow \arg\min_\mu \left( \underbrace{\mathbb{E}_{Q\sim\mu}\left[R_N(\mu)\right]}_{\text{Bellman consistency}} + \underbrace{\sqrt{\frac{D_{KL}(\mu||\mu_0)}{N}}}_{\text{conservative value update}} \right.$$

$$\left. - \xi \underbrace{\mathbb{E}_{Q\sim\mu}\left[\mathbb{E}_{s\sim p_\pi}\left[\text{var}_{s'\sim P}\left[\mathbb{E}_{a'\sim\pi}\left[Q(s',a')\right]\right]\right]\right]}_{\text{exploration}} \right)$$

after few simplifications on the bound that do not make a practical influence on the outcome. This objective comprises three terms with complementary and interpretable contributions. The first term, *Bellman consistency*, encourages accurate policy evaluation. The second term *conservative value update* mitigates the overfitting risk due to estimation errors. Although we do not explore it in this work, another plausible feature of this training objective is that it allows to build conservative policy iteration algorithms by updating the prior of an episode with the posterior of the previous episode as in Peters et al. (2010); Schulman et al. (2015, 2017). The expected variance term maximizes the expected variance of the next state, hence promotes *exploration*. The emergence of this term from first principles, unlike the manually added maximum entropy term in (1) is remarkable. We tune the contribution of this term to the loss by a new hyperparameter $\xi \in [0,1]$. As this term is always positive, any choice of $\xi$ preserves the validity of the corresponding PAC Bayes bound.

### 3.2. Implementation

Initially, we model the posterior distribution on our critic as a neural network with normal distributed penultimate layer parameters: $w_i \sim \mathcal{N}(w_i|m_i, v_i)$, $Q|s, a = \sum_{i=1}^K w_i \phi_i(s,a) + b$, for weights $w_1, \ldots, w_k$ and deterministic bias $b$. This random process is equivalent to $Q|s, a \sim \mathcal{N}(Q|m_p, v_p)$ with $m_p(s,a) := \sum_{i=1}^K m_i \phi_i(s,a) + b, v_p(s,a) := \sum_{i=1}^K v_i \phi_i^2(s,a)$. Hence, $\mu|(s,a) := \mathcal{N}(m_p(s,a), v_p(s,a)), \forall (s,a) \in \mathcal{S} \times \mathcal{A}$. We define the corresponding priors on the penultimate layer weights as a standard normal distribution. Hence we have

$$R_N(\mu) = \frac{1}{N} \sum_{(s,a,r,s')} \left(r + \gamma \texttt{stop\_grad}(Q)(s') - m_p\right)^2 + v_p.$$

Here, $\texttt{stop\_grad}(\cdot)$. Unlike the common practice that trains two critics and uses their minimum as the target estimator (Fujimoto et al., 2018), we train a single critic that mitigates the overestimation bias thanks to the conservatism provided by the PAC Bayes bound. The KL divergence term is analytically tractable as well. The only remaining term is the expected variance of the value of the next state. An exact calculation requires access to the state transition model and its bias-free estimation requires multiple Monte Carlo samples taken from each specific current state. As neither of the two are feasible, we approximate this quantity by the variance of the critic averaged across the samples:

$$\mathbb{E}_{Q\sim\mu}\left[\mathbb{E}_{s\sim p_\pi}\left[\text{var}_{s'\sim P}\left[\mathbb{E}_{a'\sim\pi}\left[Q(s',a')\right]\right]\right]\right] \approx \frac{1}{N} \sum_{(s,a,r,s')\in D_N} \mathbb{E}_{a'\sim\pi}\left[Q(s',a')\right].$$

The final critic training objective is then, with $\mathbf{m} = \{m_1, \ldots, m_K\}$ and $\mathbf{v} = \{v_1, \ldots, v_k\}$,

$$L(\mathbf{m}, \mathbf{v}) := \frac{1}{N} \sum_{(s,a,r,s')} \left[ (r + \gamma \texttt{stop\_grad}(Q)(s') - m_p(s,a))^2 + v_p(s,a) - \mathbb{E}_{a' \sim \pi} \left[ v_p(s',a') \right] \right]$$
$$+ \sqrt{\frac{\sum_{i=1}^K D_{\mathrm{KL}}(\mathcal{N}(Q|m_i, v_i) || \mathcal{N}(Q|0,1))}{N}}.$$

We follow the actor training scheme of SAC introduced in (2) with the only difference being that the $Q$ function is sampled each time it is evaluated, i.e.,

$$\arg\max_\pi \frac{1}{N} \sum_{(s,a) \in D_N} \mathbb{E}_{a_i \sim \pi, \epsilon \sim \mathcal{N}(0,1)} \left[ m_p(s,a) + \epsilon \sqrt{v_p(s,a)} - \alpha \log \pi(a_i|s_i) \right].$$

**Critic-guided optimal action search by multiple shooting.** Supplementing the random actor of SAC with a random critic has interesting benefits while choosing actions at the time of real environment interaction. When at state $s$, the agent simply takes an arbitrary number of samples from the actor $a_1, \ldots, a_S | s \sim \pi$, evaluates each with samples taken from the critic distribution $\{Q_i | a_i, s \sim \mu : i = 1, \ldots, S\}$ and acts with the action $a_*$ that maximizes the set of sampled $Q$'s. Let us denote this policy as $\pi_S$. This multiple-shooting approach both accelerates the search of the optimal action and fosters exploration by introducing an additional perturbation factor. While model-guided random search has widespread use in the model-based reinforcement learning literature (Chua et al., 2018; Hafner et al., 2019b,a; Levine and Koltun, 2013), it is a greatly overlooked opportunity in the realm of model-free reinforcement learning. This is possibly due to the commonplace adoption of deterministic critic networks, which are viewed as being under the overestimation bias of the Jensen gap (Van Hasselt et al., 2016). Their guidance at the time of action might have been thought to further increase the risk of over-exploitation.

**Convergence properties.** A single shooting version of PAC4SAC, i.e., $S = 1$, satisfies the same convergence conditions as given in Haarnoja et al. (2018, Lemma 1 & 2, Theorem 1), as their proofs apply to any value approximator. For $S > 1$, we have an algorithm with a critic that evaluates $\pi_S$ but an actor that improves $\pi$, i.e., updates the actor assuming $S = 1$. Convergence holds under such mismatch between policies assumed during policy evaluation and improvement by redefining the soft Bellman backup operator as $T_{\pi_S} Q_S(s,a) := r(s,a) + \mathbb{E}_{s' \sim P, a' \sim \pi} [\gamma Q_S(s',a') - \alpha \log \pi(a'|s')]$ highlighting the nuance that $Q_S$ evaluates $\pi_S$, although the current action is taken with respect to $\pi$. Applying the Bellman backup operator as in (3) and redefining the reward function as $r_\pi(s,a) := r(s,a) - \mathbb{E}_{s' \sim P, a' \sim \pi} [\log \pi(a'|s')]$, we match the conditions of the the classical policy evaluation proof of Sutton and Barto (2018), which was adopted in Haarnoja et al. (2018)'s Lemma 1. Our algorithm satisfies the following policy improvement theorem.

**Theorem 2** *The update rule*

$$\pi' := \arg\min_\pi \mathbb{E}_{s \sim P} \left[ D_{KL} \left( \pi(\cdot|s) \middle\| \frac{\exp(Q_S(s,\cdot)/\alpha)}{\int \exp(Q_S(s,\bar{a})/\alpha) \pi(\bar{a}|s) d\bar{a}} \right) \right]$$

*satisfies $Q'_S \geq Q_S$ for any $(s,a) \in \mathcal{S} \times \mathcal{A}$ and any sample count $S > 0$ where $Q'_S$ corresponds to the value of the new policy $\pi'$.*

Table 1: Our PAC4SAC brings a consistent improvement in online reinforcement learning performance in terms of both sample efficiency and cumulative regret in four continuous control tasks with varying state and action dimensionalities.

| | Cartpole Swingup $(d_s = 5, d_a = 1)$ $(r_{\text{limit}} = 850)$ | Half Cheetah $(d_s = 17, d_a = 6)$ $(r_{\text{limit}} = 2000)$ | Ant $(d_s = 111, d_a = 11)$ $(r_{\text{limit}} = 2500)$ | Humanoid $(d_s = 376, d_a = 17)$ $(r_{\text{limit}} = 3500)$ |
|---|---|---|---|---|
| | Cumulative Regret ($\downarrow$) $\cdot 10^3$ | | | |
| DDPG | $7.2_{\pm 1.0}$ | $317.8_{\pm 24.0}$ | $210.8_{\pm 21.2}$ | $906.2_{\pm 7.8}$ |
| SAC | $6.5_{\pm 0.3}$ | $166.8_{\pm 10.4}$ | $165.5_{\pm 17.0}$ | $539.0_{\pm 20.0}$ |
| OAC | $22.7_{\pm 1.4}$ | $213.0_{\pm 15.5}$ | $443.8_{\pm 128.3}$ | $1223.7_{\pm 44.5}$ |
| PAC4SAC (Ours) | $\mathbf{5.7}_{\pm 0.3}$ | $\mathbf{132.8}_{\pm 10.8}$ | $\mathbf{113.3}_{\pm 10.9}$ | $\mathbf{528.8}_{\pm 36.5}$ |
| | Number of Episodes Until Task Solved ($\downarrow$) | | | |
| Max training episodes ($E_{\text{max}}$) | 40 | 250 | 500 | 500 |
| DDPG | $34.2_{\pm 3.8}$ | $250.0_{\pm 0.0}$ | $298.6_{\pm 56.4}$ | $500.0_{\pm 0.0}$ |
| SAC | $24.4_{\pm 5.7}$ | $250.0_{\pm 0.0}$ | $302.0_{\pm 44.8}$ | $482.2_{\pm 15.9}$ |
| OAC | $40.0_{\pm 0.0}$ | $250.0_{\pm 0.0}$ | $315.0_{\pm 82.6}$ | $500.0_{\pm 0.0}$ |
| PAC4SAC (Ours) | $\mathbf{22.0}_{\pm 5.1}$ | $\mathbf{223.6}_{\pm 14.6}$ | $\mathbf{146.4}_{\pm 12.3}$ | $\mathbf{473.8}_{\pm 18.5}$ |

*mean $\pm$ standard deviation over five random seeds. Lowest mean is marked* **bold**

The proof given in Appendix A is an adaptation of Haarnoja et al. (2018, Lemma 2) to the case that $\pi_{k+1}$ improves not only $Q$ but also $Q_S$. Putting this theorem together with the policy evaluation proof sketched above in the same way as Haarnoja et al. (2018, Theorem 1), the algorithm is guaranteed to converge to the optimal policy.

## 4. Experiments

**Performance metrics.** We compare the performance of PAC4SAC to the state of the art with respect to: *(i) Number of episodes until task solved*: $\min(E_{\text{solved}}, E_{\text{max}})$ is the minimum between the first episode where a model exceeds $r_{\text{limit}}$ in five consecutive episodes and the maximum number of training episodes $E_{\text{max}}$. It measures how quickly an agent solves the task. *(ii) Cumulative Regret* defined as $\sum_{i=1}^{E_{\text{solved}}}(r_{\text{limit}} - r_i)$, where $r_{\text{limit}}$ is a cumulative reward limit for an episode to accomplish the task in the environment and $r_i$ is the cumulative reward for episode $i$. It measures how efficiently the agent solves the task.

**Experimental details.** We report experiments in four continuous state and action space environments with varying levels of difficulty: Cartpole Swingup, Half Cheetah, Ant, and Humanoid. We use the PyBullet physics engine (Coumans and Bai, 2016–2019) under the OpenAI Gym environment (Brockman et al., 2016) with PyBullet Gymperium library (Ellenberger, 2018–2019). While our method is applicable to any actor-critic algorithm, we choose SAC as our base model due to its wide reception as the state-of-the-art. We compare against DDPG (Lillicrap et al., 2016) as a representative alternative actor-critic design.

We train all algorithms with step counts proportional to the state and action space dimensionalities of environments: 40000 for Cartpole Swingup, 250000 for Half Cheetah, and 500000 for Ant and Humanoid. Having observed no significant improvement afterwards in preliminary trials, we terminate training at these step counts to keep the cumulative regret scores more comparable. We select $r_{\text{limit}}$ as 850 for Cartpole Swingup, 2000 for Half Cheetah, 2500 for Ant, and 3500 for Humanoid according to the final and best cumulative rewards of the models. We report all results for five experiment repetitions. We take 500

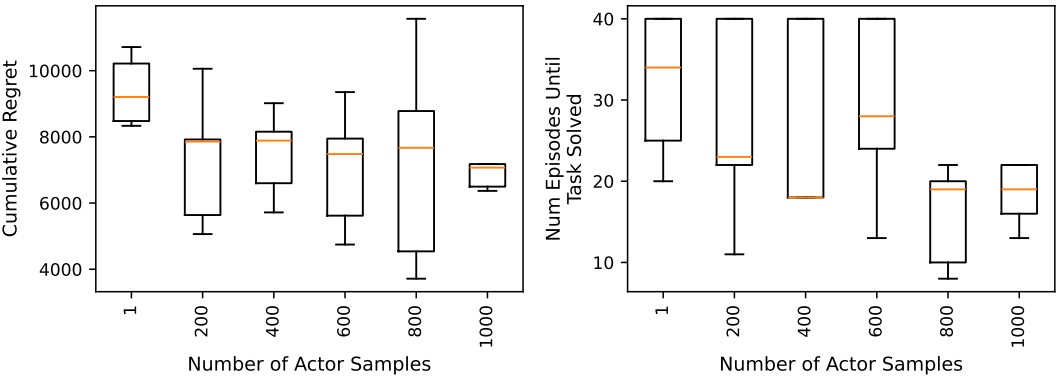

Figure 2: The effect of critic-guided random optimal action search (multiple shooting) on the performance of our PAC4SAC algorithm demonstrated on the cartpole swingup environment. Taking more samples reduces cumulative regret *(left panel)* and improves sample efficiency *(right panel)*.

Table 2: Performance contribution of the critic training loss terms. When used alone, the Bellman consistency term yields high cumulative regret. Both the conservative update and exploration terms bring performance improvements. The complete critic loss reaches the lowest cumulative regret and brings improved sample efficiency.

| Bellman consistency | conservative value update | exploration | Cartpole Swingup | | Half Cheetah | |
|:---:|:---:|:---:|:---:|:---:|:---:|:---:|
| | | | Cumulative Regret $(\cdot 10^3)$ | Num Episodes Until Task Solved | Cumulative Regret $(\cdot 10^3)$ | Num Episodes Until Task Solved |
| ✓ | ✗ | ✗ | $6.5_{\pm 1.5}$ | $24.6_{\pm 5.9}$ | $182.2_{\pm 14.4}$ | $250.0_{\pm 0.0}$ |
| ✓ | ✓ | ✗ | $5.9_{\pm 0.3}$ | $25.4_{\pm 3.4}$ | $141.6_{\pm 21.5}$ | $\mathbf{205.2_{\pm 21.1}}$ |
| ✓ | ✗ | ✓ | $7.1_{\pm 1.0}$ | $\mathbf{17.0_{\pm 2.0}}$ | $153.0_{\pm 17.3}$ | $211.6_{\pm 21.1}$ |
| ✓ | ✓ | ✓ | $\mathbf{5.7_{\pm .3}}$ | $22.0_{\pm 5.1}$ | $\mathbf{132.8_{\pm 10.8}}$ | $223.6_{\pm 14.6}$ |

*mean ± standard deviation over five random seeds. Lowest mean is marked **bold***

action samples for PAC4SAC in all experiments. We give further details of the experiments in Appendix B. Our results can be replicated using our repository: https://github.com/adinlab/PAC4SAC. We present our main results in Table 1. The table demonstrates a consistent performance improvement in favor of our PAC4SAC in all four environments in terms of both sample efficiency and cumulative regret.

**Ablation Study.** We evaluate the effect of individual loss terms on the performance in the Cartpole Swingup and Half Cheetah environments in Table 2. Used alone, the Bellman consistency term learns faster but with higher cumulative regret. All three terms are required to minimize cumulative regret. We also observe in Figure 2 that PAC4SAC learns faster and incurs less cumulative regret when it takes more actor samples.

## 5. Related work

**Actor-critic algorithms.** The Deep Deterministic Policy Gradient (DDPG) algorithm (Lillicrap et al., 2016) is a pioneering work in adapting actor-critic algorithms to deep learning. It trains a critic to minimize the Bellman error and a deterministic actor to maximize the critic, following a simplified case of the policy gradient theorem. Fujimoto et al. (2018)'s Twin Delayed Deep Deterministic Policy Gradient (TD3) algorithm improves the robustness of DDPG by adopting twin critic networks backed up by Polyak averaging updated target copies. Trust region algorithms such as Relative Entropy Policy Search (Peters et al., 2010), Trust Region Policy Optimization (Schulman et al., 2015), and PPO (Schulman et al., 2017) aim to explore while guaranteeing monotonic expected return improvement and maintaining training stability by restricting the policy updates via a KL divergence penalty between the policy densities before and after a parameter update. Han and Sung (2021) provide sample efficient exploration by exploiting samples in reply buffer and introduce entropy regularization framework for off policy setup that maximise the entropy of weighted sum of policy action distribution. Optimistic actor critic (OAC) (Ciosek et al., 2019) introduces a sample efficient algorithm which approximates a lower and upper confidence bound on the value function and address the pessimistic underexploration and directionally uninformed problem in actor-critic methods. Pan et al. (2020) assess overestimation and underestimation problem for actor-critic approaches in continuous control setup and improve both of them by using Boltzmann softmax operator for value function estimation. Moskovitz et al. (2021) study using the pessimistic value updates to overcome function approximation errors, and provide an estimation framework which switch online between optimistic and pessimistic value learning. Antos et al. (2008) propose a variance reduced critic estimation method which finds near-optimal policy using a Vapnik-Chervonenkis crossing dimension technique in order to control the influence of variance term as a penalty factor. Finally, Lee et al. (2019) enhance the generalization ability of RL agents by incorporating a randomized network that applies random perturbations to input observations which induces robustness to the policy-gradient algorithm by encouraging exploration.

**Maximum entropy RL.** Incorporation of the entropy of the policy distribution into the learning objective finds its roots in inverse reinforcement learning (Ziebart et al., 2008), which maintained its use also in modern deep inverse reinforcement learning applications (Wulfmeier et al., 2015). The soft Q-learning algorithm (Haarnoja et al., 2017) uses the same idea to improve exploration in forward reinforcement learning. SAC (Haarnoja et al., 2018) extends the applicability of the framework to the off-policy setup by also significantly improving its stability and efficiency thanks to an actor training scheme provably consistent with a maximum entropy trained critic network. Among optimistic exploration algorithms, Seo et al. (2021) provide a sample efficient exploration method for high-dimensional observation spaces, which estimates the state entropy using k-nearest neighbors in a low-dimensional embedding space. Another exploration method for high-dimensional environments with sparse rewards presented by Zhang et al. (2021) provides a sample efficient exploration strategy by maximizing deviation from explored areas.

**PAC Bayesian learning.** Introduced conceptually by Shawe-Taylor and Williamson (1997), PAC Bayesian analysis has been used by McAllester (1999, 2003) for stochastic

model selection and its tightness has been improved by Seeger (2002) with application to Gaussian Processes. The use of PAC Bayesian bounds to training models while maintaining performance guarantees has raised attention since the pioneer work of Dziugaite and Roy (2017). Reeb et al. (2018) show how PAC Bayesian learning can be extended to hyperparameter tuning. The first work to develop a PAC Bayesian bound for reinforcement learning is Fard and Pineau (2010) who later extended it to continuous state spaces (Fard et al., 2012). PAC Bayesian bounds start to be used in classical control problems for policy search (Veer and Majumdar, 2021), as well as for knowledge transfer (Majumdar and Goldstein, 2018; Farid and Majumdar, 2021). There is no work prior to ours that employs PAC Bayesian bounds for policy evaluation as part of an actor-critic algorithm.

**Distributional RL.**  Distributional reinforcement learning (Bellemare et al., 2017) focuses on the modeling distributions over returns, i.e., focuses on aleatoric uncertainty that is inherent to the system, whereas our approach models epistemic uncertainty, i.e., uncertainty that can be reduced by an increasing amount of data. See Bellemare et al. (2023) for a recent textbook introduction to distributional RL.

## 6. Discussion, Broader Impact, and Limitations

Our results demonstrate strong evidence in favor of the benefits of using the PAC Bayesian theory as a guideline for improving the performance of the actor-critic algorithms. Despite the demonstrated empirical benefits of our approach, the tightness of the PAC Bayesian bound we used deserves dedicated investigation. We adopt the classical McAllester bound due to its convenience. There may however be alternative approaches such as second-order bound (Masegosa et al., 2020) that may leverage from the learned variance estimate of the critic distribution. The sample efficiency improvement attained thanks to a PAC Bayes trained critic may be amplified even further by extending our findings to model-based and multi-step bootstrapping setups.

The strong empirical results of this work encourages exploration of venues beyond reinforcement learning, where the PAC Bayesian theory may be useful for training loss design. For instance, diffusion models (Ho et al., 2020) may derive PAC Bayesian loss functions to improve the notoriously unstable training schemes of deep generative models by semi-informative priors.

While our PAC4SAC shows consistent improvement over existing methods, it introduces additional hyperparameters such as the variance regularization coefficient, the prior distribution, and the number of shootings. Moreover, the need for multiple shooting at the action selection time increases computational complexity linear to the number of shootings. Reusing the same minibatch sample twice in expected variance calculation accelerates computation but is likely to induce bias to the estimator. The probabilistic layers of the critic network have the additional variance parameters to be learned.

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

# Appendix

## Appendix A.  Proof of Theorem 2

Denote $V_S(s) = \mathbb{E}_{a \sim \pi}\left[Q_{\pi_S}(s,a) - \alpha \log \pi(a|s)\right]$ and the optimization objective

$$J(\pi) := \mathbb{E}_{s \sim P}\left[D_{KL}\left(\pi(\cdot|s)\middle\|\frac{\exp(Q_S(s,\cdot)/\alpha)}{\int \exp(Q_S(s,\bar{a})/\alpha)\pi(\bar{a}|s)d_{\bar{a}}}\right)\right]$$
$$= \mathbb{E}_{s \sim S, a \sim \pi}\left[\log \pi(a|s) - Q_S(s,a)\right] + \text{const.}$$

Since $J(\pi') \leq J(\pi_S)$, we have

$$\mathbb{E}_{s \sim P, a \sim \pi'}\left[Q_S(s,a) - \log \pi'(a|s)\right] \geq \mathbb{E}_{s \sim P, a \sim \pi}\left[Q(s,a) - \log \pi(a|s)\right] = V_S(s).$$

Taking $S$ action samples $\{a^{(1)}, \ldots, a^{(S)}\} \sim \pi$ and $S$ value function samples evaluated at these actions $\{Q_s^{(1)}(s, a^{(1)}), \ldots, Q_S^{(S)}(s, a^{(S)})\} \sim \mu$ and choosing the sample $a^*$ corresponding to the maximum of these function evaluations $Q_S^*(s, a^*)$ we will have

$$\mathbb{E}_{s \sim P, a \sim \pi', Q_S \sim \mu}\left[Q_S(s,a) - \log \pi'(a|s)\right] \leq \mathbb{E}_{s \sim P}\left[Q_S^*(s, a^*) - \mathbb{E}_{\pi'(a|s)}\left[\log \pi'(a|s)\right]\right].$$

Plugging this inequality into the Bellman equation and expanding it recursively, we get the desired result:

$$\begin{aligned}
Q_S(s,a) &= r(s,a) + \gamma \mathbb{E}_{s' \sim P}\left[V_S(s)\right] \\
&\leq r(s,a) + \gamma \mathbb{E}_{s' \sim P, a' \sim \pi'}\left[Q_S(s', a') - \log \pi'(a|s)\right] \\
&\leq \mathbb{E}_{s' \sim P}\left[r(s,a) + \gamma \mathbb{E}_{s' \sim P}\left[Q_S(s', a'^*) - \mathbb{E}_{a' \sim \pi'}\left[\log \pi'(a|s)\right]\right]\right] \\
&\vdots \\
&\leq Q_S'(s,a) \qquad \blacksquare
\end{aligned}$$

## Appendix B.  Experimental details

We implement all experiments with the PyTorch (Paszke et al., 2019) version 1.13.1.

**Environments.**    For experiments, we use PyBullet Gymperium library (Towers et al., 2024). We choose the environment handles `InvertedPendulumSwingupPyBulletEnv-v0` for Cartpole Swingup, `HalfCheetahMuJoCoEnv-v0` for Half Cheetah, `AntMuJoCoEnv-v0` for Ant, and `HumanoidMuJoCoEnv-v0` for Humanoid.

**Hyper-parameters.**    We use an Adam optimizer (Kingma and Ba, 2014) with a learning rate of 0.001 for all architectures. We set the length of experience replay as 25000 and batch size as 32. For PAC4SAC we set the regularization parameter to $\xi = 0.01$ and $\alpha = 0.2$.

**Architectures.**    We use the same architecture for all the environments and models in comparison. There are two main architectures: actor and critic, details of the architectures are provided in Table 3.

Table 3: Architecture details based on environments. The `SquashedGaussian` module implements a Gaussian head that uses the first $d_a$ of its inputs as the mean and the second $d_a$ as the variance of a heteroscedastic normal distribution. The `GaussLinear` module implements a fully connected layer with weights that follow independent normal distributions.

| ACTOR | | CRITIC | |
|---|---|---|---|
| SAC | | SAC | |
| PAC4SAC | DDPG | | PAC4SAC |
| OAC | | | OAC |
| | | DDPG | |
| n/a | n/a | | |
| `Linear($d_s$, 256)` | | `Linear($d_s$ + $d_a$, 256)` | |
| `LayerNorm(256)` | | `LayerNorm(256)` | |
| `Silu()` | | `Silu()` | |
| `Linear(256, 256)` | | `Linear(256, 256)` | |
| `LayerNorm(256)` | | `LayerNorm(256)` | |
| `Silu()` | | `Silu()` | |
| `Linear(256, 256)` | | `Linear(256, 256)` | |
| `LayerNorm(256)` | | `LayerNorm(256)` | |
| `Silu()` | | `Silu()` | |
| `Linear(256, 2×$d_a$, $\theta$)` | `Linear(256, $d_a$)` | `Linear(256, 1)` | `GaussLinear(256, 1)` |
| `SquashedGaussian(2×$d_a$, $d_a$)` | `Tanh()` | | |

