# OpenReview forum: "PAC-Bayesian Soft Actor-Critic Learning"
_approximateinference.org/AABI/2024/Symposium_Archival_Track — AABI 2024 - Archival Track_

### Official Review · Reviewer_KW5d · 2024-04-23
**seem sound**

**Rating:** 6
**Confidence:** 2

**Review:**

This study advocates for employing a PAC-Bayesian empirical bound as a training criterion for Soft Actor-Critic, an algorithm situated within the domain of off-policy actor-critic methods. The proposed contribution is both robust and intriguing, accompanied by comprehensive implementation details and comparative numerical evaluations with alternative approaches.

Here are some minor comments:
1) to make paper more readable, please explain what is 'cartple swingup', 'half cheetad', 'ant', 'humanoid'. Do not assume that every know what you are doing.
2) the notation of $ \mathbb{V} $ is not familiar for me!
3) equation (7) has an typo "]\|_1" . equation (8) has an typo "\|".
4) You need to define what is the bold font in the caption of tables mean? (Do not assume that every know what you are doing.)
5) Why "Prior Art" section is put in Section 4. Experiments? It is not logical for me. It should be in Section 1 or 2.

---

### Official Review · Reviewer_53Gi · 2024-04-23
**PAC-Bayesian Soft Actor-Critic Learning**

**Rating:** 6
**Confidence:** 2

**Review:**

This paper develops methods to improve soft actor critic arguments by employing the PAC Bayesian as the training objective for the critic.
Experimental studies compare the newly proposed algorithm PAC4SAC with 3 other algorithms on 4 datasets.
This paper is an unusual choice for AABI as it is not about Bayesian inference and approximation, but rather about statistical learning theory and PAC Bayesian theory.  That said the application to reinforcement learning appears to be original and the paper is consistently well presented.

It is beyond my expertise to provide a detailed critique of the theory, the novelty of the contributions and the literature review, although as mentioned the paper is well presented.

I find Table 1. confusing.  It seems that PAC4SAC is the worst algorithm in terms of cumulative regret on Half Cheetah, second worst on Ant and worst on Humanoid.  The formatting makes it difficult to tell which number is higher and the bolding isn’t applied to the lowest number.  The text claims there is a consistent improvement, so perhaps I am missing something important here.  I do think the authors should address this apparent inconsistency.


Minor comments
Equation 11
exploration -> {\rm exploration}

---

### Official Review · Reviewer_4wy6 · 2024-04-24
**May have potential but violates double-blind rules**

**Rating:** 3
**Confidence:** 4

**Review:**

## Writing Quality
The writing quality is at a mixed level. On the one hand, clarity in important concepts and explanations of theorems are appreciated. On the other hand, there are typos across the paper that needs to be addressed.

- Page 3, round box, "reduce the underestimation reduced the underestimation"
- Page 4, round box, "the value this term is not known"

## Originality and Significance
The idea of combining PAC Bayesian learning with reinforcement learning is relatively new but PAC reinforcement learning is not new. It would be better if the author can provide a background analysis for PAC RL and posit this paper against those literature. For example, PAC model free reinforcement learning: https://cseweb.ucsd.edu/~ewiewior/06efficient.pdf

Given the current version, I am not sure whether there is enough novelty.

## Pros
- The analysis is thorough. Not only theoretical bounds are derived but also practical implementations are discussed.
- Experiment results look good.

## Cons
- If thm 1 is not realistic at all, why not start with the practical bound directly?
- There might be too much technical details in the paper without sufficient explanations and examples. Quoting other established bounds is ok but we need to see more explanations and examples in why using the concepts.

---

### Author Rebuttal · Authors · 2024-05-07

We thank all reviewers for their reviews and thorough reading of our submission and answer each of the comments in turn below the respective reviews.

Please let us know in case you have any remaining comments and questions you would like to discuss.

---

### Meta-Review · Area_Chair_8bpf · 2024-05-25

**Recommendation:** Accept (Poster)
**Confidence:** 3

**Metareview:**

Reviewers broadly agree that this is interesting work, and provides contribution to PAC Bayes for RL. The biggest issue for me seems to be whether it is relevant for this AABI venue, as PAC-Bayes is not strictly an approximate Bayes technique by itself, as pointed out by reviewer 53Gi. However, I believe that it is close enough to be worth connecting these communities to see further links between the two, which is a major purpose of AABI. Therefore I recommend accept. However, I very strongly recommend that the authors re-write /add parts to the Introduction that point out connections between the Bayesian crowd and PAC-Bayes / Generalised Bayesian Learning crowd, so that it is more understandable to this venue's audience.

---

### Decision · Program_Chairs · 2024-05-27

Accept